# The Impact of Chronic Stress on Behavior and Body Mass in New Animal Models

**DOI:** 10.3390/brainsci13101492

**Published:** 2023-10-22

**Authors:** Anisia Iuliana Alexa, Carmen Lăcrămioara Zamfir, Camelia Margareta Bogdănici, Andra Oancea, Alexandra Maștaleru, Irina Mihaela Abdulan, Daniel Constantin Brănișteanu, Alin Ciobîcă, Miruna Balmuș, Teodora Stratulat-Alexa, Roxana Elena Ciuntu, Florentina Severin, Mădălina Mocanu, Maria Magdalena Leon

**Affiliations:** 1Department of Surgery II, Discipline of Ophthalmology, “Grigore T. Popa” University of Medicine and Pharmacy, 700115 Iasi, Romania; alexa_anisia@yahoo.com (A.I.A.); camelia.bogdanici@umfiasi.ro (C.M.B.); dbranisteanu@yahoo.com (D.C.B.); roxana-elena.ciuntu@umfiasi.ro (R.E.C.); 2Department of Morpho-Funcțional Sciences I, Discipline of Histology, “Grigore T. Popa” University of Medicine and Pharmacy, 700115 Iasi, Romania; carmen.zamfir@umfiasi.ro; 3Department of Medical Specialties I, “Grigore T. Popa” University of Medicine and Pharmacy, 700115 Iasi, Romania; alexandra.mastaleru@gmail.com (A.M.); irina.abdulan@yahoo.com (I.M.A.); leon_mariamagdalena@yahoo.com (M.M.L.); 4Department of Biology, Faculty of Biology, “Alexandru Ioan Cuza” University Iasi, 700505 Iasi, Romania; 5Center of Biomedical Research, Romanian Academy, 700506 Iasi, Romania; 6Academy of Romanian Scientists, Str Splaiul Independentei no. 54, Sector 5, 050094 Bucharest, Romania; 7Preclinical Department, Apollonia University, Pacurari Street 11, 700511 Iasi, Romania; 8Department of Exact Sciences and Natural Sciences, Institute of Interdisciplinary Research, “Alexandru Ioan Cuza” University of Iasi, Alexandru Lapusneanu Street, No. 26, 700057 Iasi, Romania; balmus.ioanamiruna@yahoo.com; 9Department of Medical Oncology, Discipline of Oncology-Radiation Therapy, “Grigore T. Popa” University of Medicine and Pharmacy, 700115 Iasi, Romania; teodora.alexa-stratulat@umfiasi.ro; 10Department of Surgery II, Discipline of Oto Rhino Laryngology, “Grigore T. Popa” University of Medicine and Pharmacy, 700115 Iasi, Romania; florentina.severin@gmail.com; 11Department of Medical Health III, Discipline of Dermatology, “Grigore T. Popa” University of Medicine and Pharmacy, 700115 Iasi, Romania

**Keywords:** stress, open field, running, swimming, light exposure variation

## Abstract

(1) Background: Exposure to different sources of stress can have a significant effect on both psychological and physical processes. (2) Methods: The study took place over a period of 34 days and included a total of 40 animals. Regarding the exposure to chronic stressors, we opted for physiological, non-invasive stressors, e.g., running, swimming, and changes in the intensity of light. An unforeseen stress batch was also created that alternated all these stress factors. The animals were divided into five experimental groups, each consisting of eight individuals. In the context of conducting the open field test for behavioral assessment before and after stress exposure, we aimed to investigate the impact of stress exposure on the affective traits of the animals. We also monitored body mass every two days. (3) Results: The control group exhibited an average increase in weight of approximately 30%. The groups exposed to stress factors showed slower growth rates, the lowest being the running group, recording a rate of 20.55%, and the unpredictable stress group at 24.02%. The anxious behavior intensified in the group with unforeseen stress, in the one with light variations, and in the running group. (4) Conclusions: Our research validates the animal model of intermittent light exposure during the dark phase as a novel method of inducing stress. The modification of some anxiety parameters was observed; they vary according to the type of stress. Body mass was found to increase in all groups, especially in the sedentary groups, likely due to the absence of cognitive, spatial, and social stimuli except for cohabitation.

## 1. Introduction

Long-term emotional stress causes both temporary and permanent changes in behavior and physiological processes, which are crucial factors in the development of many psychiatric diseases. Such emotional triggers significantly contribute to the pathophysiology of both depressive and anxiety conditions. They are among the most significant sources of stress in everyday existence, particularly for those in the lower hierarchy [1]. Thus, studying long-term emotional stress and its effects has an increasing impact on our everyday life.

Studying long-term emotional stress involves knowing anxiety-related disorders that can be broadly classified according to symptoms and susceptibility to pharmaceutical and psychological treatment. Procedures are developed to induce conditioned behaviors or ethologically appropriate conflict. Numerous behavioral testing procedures were devised to simulate human pathological anxiety in rodents. 

The aim of our study was to validate new animal models of chronic stress and its impact on behavioral parameters and body mass variation and to compare them with pre-existing models and with a control group. Our objectives were to validate these models from a behavioral point of view through the open field test, which allows us to methodically evaluate rats’ exploratory behavior in novel environments, their general locomotion, and their anxiety-related behavior. We propose that the animal models simulate the chronic daily stress of people such as athletes who perform sports or the activities of people who have jobs that require the performance of unforeseen activities during the inactive period of the day or the situation of people who are constantly exposed to changes in the intensity of light and circadian rhythm. Unforeseen stressful events or expected adverse events induced by physical or environmental factors are used in several of these models to elicit a frightened reaction. In rats, these reactions are seen as adequate for the circumstances at hand; in comparison, anxiety in humans is considered a maladaptive or pathological reaction to the current circumstances [2].

Animal models have their own limitations due to neurological differences between species, but experimental observations can be extrapolated to better understand the impact of each type of stress on behavioral parameter disturbances and body mass variations.

## 2. Materials and Methods

### 2.1. Animals

In this study, we used white laboratory rats of the Wistar breed (*Rattus norvegicus*) as the experimental subjects. The rats selected for the study were exclusively male, with a body weight ranging from 250 to 300 g. They were obtained from the “Victor Babes” National Institute of Research and Development in Bucharest. The rats were accompanied by a certificate that confirmed their hygiene and compliance with standards. 

The animals in this study were provided with appropriate conditions for habituation. The laboratory maintained a constant temperature of 22 °C and a humidity level of 50%. The animals had unrestricted access to water and food, and the lighting conditions were adjusted to follow the circadian rhythm of 12 h of light and 12 h of darkness. Upon arrival at the laboratory premises, the animals were given a minimum of one week to acclimate before the start of the experiments in order to diminish the errors that can occur in testing due to the stress that can be achieved due to transport. Moreover, the rats needed to adapt to the new environment, including the food, light, humidity, and space. The animals were housed in plastic and metal mesh cages during the experiments. The cages were equipped with devices for food and water access. Initially, when the animals arrived, they were placed in plastic cages with four individuals per cage. The body masses of each animal were recorded for documentation purposes. 

### 2.2. Description of the Animal Models Used in the Study

In this study, multiple animal models were developed to stimulate different forms of physiological stress. These models were based on extensive experimental research from specialized literature that focused on stress. The rationale behind this approach stems from the understanding that physiological and psychological stress plays a significant role in various pathologies associated with the degeneration of organs.

Regarding exposure to stressors, we opted for three external, physiological, non-invasive stressors: running, swimming, and changes in the intensity of light in the animal housing. The current literature highlights the significant impact of these factors in rats. For instance, running is a common defense mechanism in the behavior of rats, triggering a robust activation of the sympathetic nervous system mechanism and the release of adrenaline [3]. Studies have demonstrated that subjecting animals to forced swim tests elicits behavior like stress and anxiety responses, even though rats are dual-habitat animals with both terrestrial and aquatic tendencies. Additionally, exposure to changes in light intensity has been found to have significant implications for the circadian rhythm of animals [4].

In this study, the validation and verification of the animal models used involved conducting a behavioral test on the animals. Previous research demonstrated that exposure to physical and psychological stress can lead to the development of anxious behavior [5]. In order to assess both general locomotor activity and anxious behavior in rats, we used the open-field test. Although there is not a consensus regarding the relevance of this test for evaluating anxiety in animals, it has been widely utilized for this purpose [6]. The maze used is an open-top cube with matte dark-colored walls with no distinct markings. It has a side length of approximately 40 cm. The lower part of the maze is divided into nine zones consisting of a central zone and eight marginal zones that are delineated by one or more walls. During the test, the animal is placed in the center of the maze, and its behaviors are observed and recorded for five minutes. The behaviors assessed during the test include the time spent in and the number of entries into the central square, the number of transitions between zones, grooming behavior (body cleansing), stretching posture, rearing posture (exploration of the upper part of the walls), freezing behavior (immobility), defecation, and urination. The manifestation of anxious behavior in this test is characterized by moments of freezing, reduced mobility, decreased time spent in the central square, anxiolytic grooming behavior, and the presence of feces and urine in the experimental area [7]. 

### 2.3. Experimental Design

The current study included a total of 40 animals. All animals underwent a physical training test on the fourth day after their arrival to evaluate their cooperation. Thus, the examination involved placing them on a treadmill for three minutes, with an average speed of one kilometer per hour. Based on their positive response to this activity, 16 animals were selected and subsequently assigned to the experimental groups involved in this type of activity. The animals were divided into five experimental groups, each consisting of eight individuals. The division of the animals into these groups was based on the specific stressor applied, as described in Table 1.

On the 6th day, the open field test was conducted for all experimental groups. The behavior tests were recorded on video and later reanalyzed to ensure a thorough analysis of the animal’s behavior. 

From the 9th day onwards, for five days, the animals in groups 1, 2, and 3 underwent training sessions specific to their assigned stressors, as shown in Table 2.

For animals from group 4, a consistent regimen was implemented to maintain a normal circadian rhythm, which includes regular cycles between light and dark phases, with 12 h of light between 7 a.m. to 7 p.m. and 12 h of darkness from 7 p.m. to 7 a.m.

Starting from the 14th day up to and including the 34th day, the animals from groups 1 and 2 were exposed to the stress factors, as shown in Table 3. 

During the same period, group 3 was exposed to different types of stress, according to the program described in Table 4.

For animals in group 3, the same stressor was not applied for two consecutive days to minimize predictability.

The treadmill used had a 0° incline and allowed for a gradual increase in running speed of 0.5 km/h. A wooden boundary device was constructed around it to ensure that the animals remained on the treadmill during the training sessions. This device measured 1.5 m in length and 40 cm in width. The front part of the device, where the animals ran, was covered by a wooden ceiling of 50 cm. The remaining 1 m length of the device was exposed to the ambient light of the room where the experiment took place. This enclosure design served the dual purpose of keeping the animals on the treadmill and encouraging them to run toward the covered area, which instinctively sheltered them. An electrical fence was not used to stimulate the animals to minimize additional stress factors that could influence the running pattern (Figure 1).

To induce stress through swimming, each rat was individually placed in a vessel with a diameter of 140 cm, a water depth of 40 cm, and a temperature of 30 °C (Figure 2). The room temperature was maintained at 25 °C using an air conditioning system. Both the room and water temperatures were monitored at the end of each experiment. To prevent the influence of alarm pheromones left behind by previously swimming animals, the water was changed after each swimming session, thus avoiding any additional stressors.

It is important to note that the stress models used in this study did not involve food or water deprivation.

On day 33, following the administration of the stressor, the animals underwent the open field test, which was recorded on video for later analysis. Subsequently, tears were collected after the subcutaneous injection of pilocarpine 1% at a dosage of 2 mL per kilogram of body weight. On day 34, after the injection of the stressor, the animals were euthanized by intraperitoneal injection of ketamine and xylazine in a 1:1 ratio.

## 3. Results

An animal modeling approach was employed to test the research hypothesis and achieve the objective of this study’s aim. Stress, regardless of its type (light, vigorous physical activity), determines behavioral changes that can be observed in various physiological and psychological disturbances of the study animal. 

A total of 40 animals were included in the study, and all of them completed the experiments.

### 3.1. The Effects of Exposure to External Stress Factors on the Behavior of Animals in the Open Field Test

In the context of conducting the open field test for behavioral assessment, we aimed to investigate the impact of stress exposure on the affective traits of the animals. Specifically, we examined the following behaviors: -degree of disinhibition, which corresponds to the amount of time spent in the central square of the maze;-general mobility, which corresponds to the number of crossings of the lateral squares;-anxious behavior, which corresponds to the frequency of stretching behavior in the horizontal plane and the duration of freezing or immobility;-anxiolytic behavior, which corresponds to the duration of fur cleaning behavior.

The behavioral assessments were conducted both before and after the entire duration of stressor exposure to account for individual variations among the animals in the study groups and minimize potential errors. Prior to the application of stress sources, statistically significant variations were observed in certain parameters, including the duration of anxiolytic behavior (*p* = 0.02) and immobility time (*p* < 0.001). 

Following the application of stress factors, additional parameters in the behavioral assessment exhibited statistical significance (*p* < 0.05) or showed visible differences within a narrower confidence interval (*p* < 0.08) (as depicted in Figure 3). These variations are explainable and will be further discussed. 

Furthermore, within the same behavioral test, other physiological behaviors can be evaluated to determine the level of emotional distress experienced by the animals, such as the number of urine traces and fecal boluses released during the test.

The data presented represent a comparison of these parameters before and after applying the stressors. The sample size was n = 8, independently of the group study. Statistical analysis using the *t*-test assuming equal variance revealed significant differences (* *p* < 0.005, ** *p* < 0.001).

These parameters can serve as indicators of the animals’ anxiety levels. Interestingly, no significant variations were observed when comparing the same parameters based on the applied stress factor. However, notable differences were observed when comparing the parameters before and after applying the stress factors (as shown in Figure 4).

### 3.2. Dynamics of Body Mass Depending on the Application of the Stress Factor

The animals’ body weights were measured every two days throughout the experiment. This parameter served as an indicator of the animal’s growth and development in relation to their exposure to stress factors over a specific period. The control group exhibited an average increase in weight of approximately 30%. In contrast, the groups exposed to stress factors showed slower growth rates, with the running group recording a rate of 20.55%, the swimming group at 26.02%, the unpredictable stress group at 24.02%, and the group exposed to changes in ambient lighting at approximately 28% (Figure 5). It is important to note that the nature, intensity, and duration of the stress exposure influenced these increases.

In Figure 6, the trend of weight over time is depicted, with consideration of the specific stress factor applied. The stages of modification in the parameters of the stressors, including the intensity and duration of the stimuli, were also recorded.

Under these experimental conditions, we observed changes in the dynamics of weight, particularly during the initial stages of the experiment. Some groups exhibited progressive variations in body mass (such as the swimming group), while other groups, such as the control or running groups, displayed uneven variations.

## 4. Discussion

The induction protocols for chronic stress in animal models are widely used in research. However, discrepancies in reported effects may indicate differences in methodology. Thus, most specialized laboratory protocols are implemented during the diurnal phase of the day, which corresponds to the resting period of the animal. In the present study, the same approach was followed for the stress-exposed groups subjected to running or swimming or unpredictable stress except for the group in which illumination was modulated during the night. This means that the rats could be exposed to stressors both during the resting periods (light phase) and the active periods (dark phase). A study conducted by Aslani et al. found that when the stressor was applied during the light phase, the rats exhibited signs of depressive and anxious behavior, as well as significant dendritic atrophy in the hippocampal neurons. However, these phenotypes were not observed when the same stressors were applied during the dark phase. These observations demonstrate that the timing of stress application has differential effects on behavioral and neurostructural phenotypes [8]. Moreover, the research carried out by Nagy et al. indicated that being exposed to chronic stress could lead to comparable changes in the microstructure of the white matter. These changes are similar to the ones observed in psychiatric disorders related to stress [9]. Moreover, the impact of stress on animals is influenced by various factors, including the timing and type of stressors [10]. When animals are exposed to repeated external stress during their inactive phase, it has a more negative impact. On the other hand, chronic psychosocial stress has a more significant harmful effect when experienced during the active phase. Thus, these results suggest that the impact of stressors is influenced not only by the circadian phase of exposure but also by the interaction between the circadian phase, the type of the stressor, and the chronicity of the stressors [11]. 

In our study, we used disruption of light intensity as a method to induce stress by alternating between light on and light off every hour during the dark cycle. This is important because laboratory rodents are nocturnal, and the significance of environmental stimuli and their perception and response may be a function of their activity periods when studying the effects of chronic stress.

Light plays a significant role in regulating the circadian rhythm, neuroendocrine system, and behavior, exerting a substantial impact on all animals’ overall health and well-being. Rhythmicity is induced in each cell through coordinated mechanisms in the brain via the hypothalamic–pituitary–adrenal axis, allowing mammals to adapt to the light/dark cycle. Thus, the circadian rhythm is genetically configured and involves coordinated feedback connections between centers that ensure the reception and integration of changes in environmental stimuli. 

In our experiment, we started with the consideration that exposure to light stimuli during the night disrupts the maintenance of the initial coordinates imposed by the circadian rhythm, altering the organism’s reactions [12]. The design of the group with exposure to light during the night was created based on the fact that many hormones and enzymes fundamental for life are secreted following a rhythmic circadian pattern in accordance with light exposure.

The rhythmic secretion of the hypothalamic–pituitary axis is influenced by numerous internal and external factors such as light exposure, sleep and wakefulness, stress, diet, age, or sex [13]. For instance, during the night, melatonin, a hormone that serves as a hormonal signal of photoperiodicity, is secreted [14]. The circadian release of melatonin is regulated by the suprachiasmatic nucleus of the hypothalamus, which acts as a 24 h oscillator, depending on the perceived lighting conditions by retinal receptors, cone, and rod cells, which subsequently transmit the stimulus through ganglion cells via the optic nerve [4,15]. Circulating melatonin levels display a diurnal variation characterized by elevated concentrations during the nocturnal period and reduced concentrations during the diurnal phase [16]. Conversely, the plasma concentration of leptin, a hormone secreted by adipocytes, exhibits a temporal pattern with an increase observed in the afternoon and a peak reached between midnight and approximately 2:30 a.m. [17].

Our experimental study utilized two distinct forms of physical exertion, described in the specialized literature as capable of inducing stress. Consistent with these findings, we found that both types, namely swimming and treadmill running, induced stress. Different treadmill running models described in the literature have been found to induce stress. Liu et al. describe a model that involves a 2-week training period during which the rats ran at an increasing speed until reaching a threshold of 1.6 km/h, followed by an increase in running time. After the two-week training period, all rats ran for 2 h at 1.6 km/h, five days a week, for eight weeks. To motivate the rats to run on the treadmill, an electric fence was placed behind it [18].

Our research also created a model of chronic stress through running. Unlike the model proposed by Liu et al., we chose not to place an electric fence to avoid introducing an additional stress factor and to minimize the potential physical trauma to the animals caused by contact with the electric fence.

Sun et al. proposed a model of stress through endurance exercise in rats. The rats were trained for four weeks on a treadmill, six days per week (Monday–Saturday), followed by an additional four weeks of high-intensity exercise. The rats ran until they could no longer maintain themselves on the treadmill due to exhaustion. Exhaustion was defined as the inability to continue running despite contact with the electric shock bar placed at the back of the treadmill. The outcomes of this four-week endurance exercise revealed notable consequences on the hepatocytes of the rats. Specifically, mitochondrial function was impaired, and oxidative stress was induced, highlighting the physiological impact of this exercise model on the cellular process [19]. However, Silva et al. performed a study to investigate the impact of a lifelong aerobic training program conducted at a moderate intensity on the aging process and the adaptative response of the liver in rats. The findings of this research indicate that this particular form of exercise has the potential to enhance mitochondrial function and mitigate disruptions in the antioxidant function that occur in the aging liver [20]. 

Current literature describes two general models for inducing stress through swimming: the forced and endurance swim tests. One variant of the forced swim test involves using a 20 × 20 × 50 cm container filled with 20 cm of water at 25 °C, allowing the rats to touch the bottom of the container with their hind paws and tail or balance on their hind legs against the opposite walls of the container. The endurance swim test involves using a 50 × 50 × 50 cm pool with a water depth of 40 cm at a temperature of 30 °C. The deeper water level restricts the animals’ ability to use their hind legs and tail to remain immobile, and the larger surface area and greater distance between walls make it more difficult for the rats to keep their heads above water [21].

The endurance swim model is more stressful than the forced swim model due to the more intense sensation of drowning experienced by the animal, and this hypothesis is supported by the greater increase in plasma corticosterone levels and behavioral tests, which reveal a higher level of anxiety in the animals. In our study, this is one of the reasons that motivated us to choose the endurance swim model as a stress model.

The temperature of water has a significant impact on the swimming performance of rats. If the water temperature reaches 42 °C or higher, the rat has a low physical performance due to hyperthermia. On the other hand, a water temperature of 20 °C or lower leads to hypothermia, resulting in reduced physical performance. However, since the rat’s body temperature is between 33 °C and 36 °C, if the water is maintained slightly cooler than the animal temperature, they can sustain a consistent temperature during exercise without negatively impacting cardiovascular parameters [22].

In our experiment, to avoid inducing additional stress factors to swim, the water temperature was maintained constant at 30 °C following the model described by Bruner and Vargas, who reported that at this water temperature, the heat exchange between the water and the animal’s body is minimal [23].

Arshadi and his colleagues studied endurance swimming and the variation in plasma glucose and cardiac antioxidant enzyme activity in diabetic rats. They proposed the following swimming model, which included a 7-day training phase where the animals swam in a round plastic container (140 cm–60 cm–45 cm, water temperature 34–36 °C), initially for 10 min, gradually increasing by 10 min each day until the rats swam for 60 min. The experiment involved swimming for 60 min per day, five days per week, for six weeks [24].

Lapmanee and his team used a different model of endurance swimming, which has been shown in previous studies to induce cardiac and skeletal muscle hypertrophy without plasma lactate accumulation. The animals swam five days per week, from Monday to Friday, for 60 min daily. The authors of this research found that their swimming model had an anxiolytic effect on the animals included in the study [25,26]. Other studies have also found that non-exhaustive aerobic exercises can have anxiolytic and antidepressant effects in healthy subjects and patients with anxiety [27].

The literature data regarding the mechanism underlying anxiety reduction through swimming, both in humans and rodents, are not fully understood. Researchers suggested that swimming can partially influence brain function, affording neuroplasticity enhancements and conferring neuroprotective properties across various neuropsychiatric pathologies [28]. The favorable cerebral outcomes associated with swimming encompass not only memory enhancement but also attenuation of oxidative stress through a reduction in reactive oxygen species, coupled with the heightened synthesis of brain-derived neurotrophic factor and neuronal growth factor [29]. Exploring the idea that diminished heat shock protein synthesis and elevated levels of circulating corticoids are indicators of stress, certain studies suggest that the anxiolytic effects of swimming might stem from its ability to regulate the expression of heat shock protein and glucocorticoid receptors in the hippocampus [30].

Another study, which aimed to compare stress biomarkers during acute exercise of known intensity in swimming and treadmill running in rats, used the maximum lactate level to determine the aerobic/anaerobic transition of the animals, and the blood concentrations of adrenocorticotropic hormone and corticosterone were used as biomarkers. The results of the study indicate that, in terms of biomarker values, swimming is a more stressful physical exercise than treadmill running, especially when exercise intensity leads to anaerobic conversion. This difference may be, at least partially, a consequence of the animals’ uncertainty in water during exhaustive efforts when they have to struggle to survive [31].

The treadmill has the advantage that exercise intensity is easily determined and can be increased by simply increasing the speed. However, among the disadvantages, the following can be mentioned: high equipment costs, the need for animal selection as not all rats run on the treadmill, the risk of injury to the animals, especially in the lower limbs, and the presence of the electric stimulus for running as a stress factor [31].

In our experiment, we decided not to use an electric fence and to enclose the treadmill so that the animals would not jump off and have no other option but to run, specifically to exclude this stress factor that could interfere with the induced stress through the chosen running model. The disadvantage was that the running speed was not increased at the same accelerated rate compared to other studies that used the electric stimulus. We closely monitored the animals throughout the running to prevent any accidents that could cause physical trauma to the animals.

Studies on swimming exercise reported that rats can adapt their physical fitness similarly to humans [32]. However, some authors criticize studies that use swimming exercises in rats, arguing that water temperature influences the level of stress and that swimming causes physical and psychological stress due to the animals’ struggle for survival. It is challenging to precisely quantify the intensity of physical effort in swimming, unlike treadmill running, but it can be reported temporally [31].

From the observations of the present study, we found that swimming exercise does not cause lower limb injuries and is less physically traumatic. Animals require constant supervision and encouragement to swim by creating water currents when the animal adopts a floating position on the water’s surface. 

In the specialized literature, various models of unpredictable stress have been described. One such model is the chronic mild stress model, which has been extensively utilized in the study of depression [33]. Unpredictable chronic stress has been used to create models for the human symptom of anhedonia, defined as the loss of interest in previously enjoyable daily activities. Studies have confirmed that chronic mild stress induces behavioral changes in rodents that resemble symptoms of depression, including reduced reactivity to rewards and other depression-like behaviors [34]. However, there have been reports of an “anomalous” behavioral profile in some animals exposed to this type of stress, which is associated with distinct neurobiological changes. These contradictory findings suggest that chronic mild stress can elicit both depressive and anomalous behavioral profiles, each linked to different patterns of neurobiological alterations [35]. Furthermore, chronic unpredictable mild stress is currently considered the most widely used method for modeling chronic stress in animals. It has been employed to investigate the effects of chronic stress on various conditions, including colonic inflammation, gastric precancerous lesions, depression, and other systemic disease [36].

Riaz et al. used two models of unpredictable stress to study anhedonia. For each model, a different stressor was applied each day for ten days, with the ten types of stressors alternating over a four-week period. The first unpredictable stress model followed this schedule for the ten-day interval, with each stressor corresponding to a specific day: 50 min in a cold room, 4 h of wet bedding, 1 h of immobilization, 50 min in a rotating cage, 15 min of isolation in a cold room, 4 h of wet bedding, 30 min of cage rotation, 5 min of swimming exposure during the dark phase of the circadian rhythm, and 45 min of isolation in a cold room for each animal. The second model involved 60 min in a rotating cage, 12 h of cage illumination during the dark phase, 3 h of darkness during the light phase of the circadian rhythm, 15 h of food and water deprivation during the dark phase, 17 h of isolation for each animal during the light phase, 3 h of darkness during the light phase, 1 h of light during the dark phase, 1 h of immobilization, 12 h of food deprivation during the dark phase, and 6 h of illumination during the night. These unpredictable stress models resulted in reduced social interaction, decreased spatial memory performance, decreased body weight gain rate, and increased latency to feed on a novel food [1]. Chronic stress has been found to play an essential role in the pathogenesis of many psychiatric disorders by inducing short and long-term changes in behavior and physiological functions [1,37]. Emotional stressors, particularly those experienced by individuals with low social hierarchy, are recognized as prominent sources of stress in human life and are closely associated with the pathophysiology of anxiety and depressive disorders [38].

Based on previous findings that patients with depression express higher levels of proinflammatory cytokines, Şahin et al. used a model of unpredictable chronic stress to induce anhedonia in animals. They found that Infliximab, a TNF-α inhibitor, reduced the level of anhedonia, suggesting that inflammation may play an important role in depression. The unpredictable stress involved the random application of nine stressors over a 56-day period, with each stressor being applied 4–5 times during the experiment. These stressors included immobilization for 4 h, cage tilt for 24 h, wet bedding for 24 h, swimming in cold water at 4 °C for 5 min, swimming in hot water at 45 °C for 5 min, cohabitation with another stressed animal for 48 h, cage movement for 10 min, tail restraint and compression for 1 min, and reversal of light/dark cycle for 24 h. Daily moderate-intensity stress over 56 days induced a depression-like state, with impairments in spatial and emotional memory [39]. Among the multiple neurochemical changes that occur in response to stress factors, excessive inflammation processes are considered to play a major role in resulting cognitive disorders [40]. One example of this phenomenon is observed in reactive astrogliosis, which occurs as a secondary response to an inflammatory insult. This process creates a positive feedback loop, leading to impaired clearance of metabolites and the accumulation of cytokines and other mediators of inflammation. Consequently, this exacerbates neuroinflammation and establishes a potential connection between the reactivity of microglia and astrocytes and the progression of neurodegenerative processes [41].

In another study, the hypothesis that olfactory function could be affected in a rat model of depression was tested using a chronic unpredictable stress model. The stress model followed a specific protocol, starting with the animals being placed in darkness for 40 min (under a cardboard box) on the first day. This was followed by 40 min in ambient light and then another 40 min in darkness. From the second day onwards, the animals were subjected to a series of stressors in a sequential manner. These stressors included spending 2 h in a clean cage without bedding, 2 h alone in a cage with no bedding, 20 min of immobilization, extending the dark phase of the circadian rhythm to 12 h, 2 h in a cage with wet bedding, 2 h on an oscillating plane, 2 h in a new cage with wet bedding and feces from an unrelated male Wistar rat, 2 h in a tilted cage at 30 °C without bedding in the dark, 2 h alone in the dark in a clean cage, extending the dark phase to 16 h, 2 h of exposure to music at 80 decibels, 2 h alone in a cage with dirty bedding and feces from unrelated Wistar males, and 20 min of immobilization [42].

In chronically stressed animals, physiological changes occur to adapt to regular challenges through sustained stimulation of the adrenal glands, which secrete catecholamines and glucocorticoids. Experimental protocols involve using various stressors to avoid habituation [43]. In the case of mild chronic stress, animals are exposed to mild stressors in an unpredictable manner over several weeks (3 to 9) [1,39,42,43].

From a behavioral perspective, it has been shown that exposure to different sources of stress can have a pronounced effect on cognitive processes, taking into account the type of stressor, its intensity, and duration. Studies suggest that impairments in memory characterize animal models exposed to stressors, affective deficits of an anxious and depressive nature, and even social deficits [44,45,46,47].

In this study, we obtained cognitive deficits similar to those reported in the scientific literature, which suggests the validity of the animal models used. Similarly, the fact that exposure to changes in artificial light exposure leads to behaviors comparable to those obtained through exposure to other types of stressors (such as running, swimming, or unpredictable stress) indicates that this type of stressor can induce changes not only in the system directly involved in perceiving these changes, but also in the whole body and, especially, in the central nervous system. These observations highlight the importance of respecting physiological needs for rest and the use of the visual system, as inadequate care can have repercussions not only on the eye as a physical part of light perception but also through its connection to the central nervous system, on other structures involved in modulating brain cognitive activities (as observed in this study, on structures that modulate stress response, anxiety, and even more profoundly, on molecular modulation mechanisms) [48,49].

In our experiment, another important observation arises from the analysis of the behavior of animals exposed to changes in light compared to the control group. Since the only difference in the design of the two study groups is the modification of the environmental parameter of ambient lighting, excluding the possibility of the influence of increased motor activity on behavior, as presented by authors of other studies focusing on the effect of running on animal models, we can observe that this stress factor can influence animal behavior in terms of the appearance of behaviors that indicate a higher level of anxiety. The behavior cannot extend to the point where animals exhibit depressive-like behaviors characterized by increased immobility durations as assessed in the open field test. Current data from the literature attribute these behavioral changes to the influence of changes in ambient lighting on the animals’ circadian rhythm and, consequently, the generation of sleep disturbances, which have been shown to generate anxious behavior, attention deficits, tendencies towards social isolation, and other behavioral impairments, as demonstrated by Pădurariu and Bains et al. in two recent animal modeling studies [50,51].

Our study, as a preliminary conclusion, provides additional evidence regarding the effect of modifying the ambient lighting schedule on the behavior of the studied animals, acting as a valid stressor with behavioral outcomes comparable to exposure to other sources of stress of different types, intensities, and formulations.

In our experiment, regarding the variation in physiological parameters in animals exposed to stressors, we observed significant variations in body mass. Regarding this parameter, the animals were divided into five groups, with no significant differences between the groups (*p* = 0.26). However, at the end of the experiment, we observed statistically significant differences between the study groups (*p* = 0.023), indicating a correlation between the type of stressor applied and the dynamics of body mass growth in the animals. This correlation is in accordance with other studies, which highlights the implications of sedentary behavior on body mass, as well as the effects of exercise, as we observed in the running and swimming groups [52].

In this study, regarding the experimental groups exposed to unpredictable stressors and changes in ambient lighting, we observed that they had intermediate growth rates compared to the control and exercise groups. The highest body mass growth rate was recorded for the control group, indicating the possibility of considering this group as an accurate sedentary model. In the same context, the lowest body mass growth rate was observed for the running group, indicating the implications of increased physical activity on growth and development. However, we cannot differentiate between the increase in muscle mass or in the adipose tissue due to the fact that the rodents were only weighed with a calibrated authorized scale. Thus, we cannot comment on whether exposure to stressors directly contributes to slowing down the growth rate of the animals or, due to the specific stressors applied, to the acquisition of adipose tissue. This represents one of the limitations of this study.

Even under these conditions, if we were to analyze the inclusion of the stress factor through changes in ambient lighting in the suite of stressors, based on the dynamics of this parameter (body mass dynamics), we observed a growth rate comparable to that obtained for the sedentary control group. Given that the animals did not move more than those in the control group, and the only difference between the two groups was the modification of the ambient lighting schedule, it can be concluded that physical activity and the alternation of stressful tasks (within the unpredictable stress group) can directly contribute to the dynamics of this physiological parameter.

The growth rate of body mass, as mentioned earlier, can indicate the tolerance of animals to the application of stressors, depending on their intensity and duration. Being a fundamental physiological parameter monitored in most types of animal modeling experiments, it can reflect both the physiological state of the animals and specific response reactions to treatments or active tasks. Thus, while in pharmacological treatments, this parameter indicates the possibility of adverse response reactions, in this case, it can be directly correlated with the animal’s development.

In this experiment, we closely monitored the variation in this parameter in order to correlate the response of animals to increased physical activity versus sedentary behavior and to classify the type of stress by comparing the effect of changes in ambient lighting with the effects of other types of stressors on behavior and physiological parameters. Therefore, by observing the variation in body mass every two days, we observed, for example, the influence of the type of stressor on the rate of body mass growth.

Our experimental study can well argue the dynamics observed for the running group, as exemplified in the presentation of the results of this study. Body mass growth started abruptly and had a progressive increase as the animals became accustomed to the intensity and duration of the stimulus. From day 24, when the second modification of the stressor parameters was made, we observed either a stagnation or a slight increase in body mass until the end of the experiment. This example closely follows the trend observed for the groups involving increased physical activity, while the group that was not subjected to exercise showed more irregular dynamics. It is worth noting that in the group exposed to changes in ambient lighting, the trends in body mass growth are discreetly similar to the dynamics observed in the sedentary control group, thus validating the similarity between the types of activities required for modeling these two groups. At the same time, although the trends are comparable, the growth rate is slightly different, with the differences discussed earlier.

Previous studies demonstrated that the disruption of the rest phase (diurnal phase) interferes with the pattern of food intake and metabolism, suggesting the disruption of circadian regulation of feeding and metabolism [53]. Another study determined that the administration of stressors during the rest period substantially reduces body mass. However, only minor fluctuations in body mass were observed when the same intervention was applied during the dark phase of the daily cycle [8].

In our experiment, by monitoring the body mass growth rate of the animals, we observed a higher and consistent growth rate in the group where the stressor (modulations of lighting) was applied during the night, compared to the groups where the stressors were applied exclusively (swimming and running group) or predominantly (unpredictable stress group) during the day, the rest period of the rats. This provides additional evidence for the variation in body mass growth rate between the groups.

## 5. Limitations and Strengths

One of the limitations of our study is the fact that we could not investigate bioimpedance regarding the rodents included in our study in order to determine the difference between the changes in their body composition (lean tissue, adipose tissue, water). Moreover, we could not quantify the ingestion of food and water and the quantity of urine and stools. 

Another limitation is provided by the anatomical and physiological differences between the species, but further research regarding the analogies between the behavioral reactions and the neuroscience of rodents participating in stress regulation can open perspective for the use of new animal models of chronic stress to test the efficacy and safety of drug treatment for treating anxiety in humans.

The strength of our study was the implementation of a novel model that induces stress by light variation, this being, as far as we know, the first study that evaluates this. Moreover, this model can be easily reproduced. In addition, we also described a model of unpredicted stress that was different from the ones in the literature because we have used different parameters.

## 6. Conclusions

In conclusion, this study provides several important aspects when compared to existing literature. Firstly, our research validates the animal model of intermittent light exposure during the dark phase as a novel method of inducing stress, which is consistent with previous studies utilizing running and swimming models. Secondly, the group exposed to stressors during the night, specifically modulations of illumination, exhibited a higher and consistent rate of weight gain compared to groups subjected to physical effort stressors applied exclusively (swimming and running groups) or primarily (unpredictable stress group) during the day, which corresponds to the resting period of rats. Thirdly, the control group, serving as an experimental sedentary group, demonstrated a more significant increase in body mass compared to other groups, likely due to the absence of cognitive, spatial, and social stimuli except for cohabitation. Moreover, through statistical correlations between behavioral parameters assessed using the open field test before and after exposure to stress factors, the modification of some anxiety parameters was observed; they vary according to the type of stress. Lastly, based on the concept of unpredictable stress found in the literature, we developed our own unanticipated stress model by alternately exposing animals from the respective group to the stress factors studied in each of the other groups (swimming, running, sudden changed in light throughout the night), making it the first study to combine these three stressors in an experimental animal model.

## Figures and Tables

**Figure 1 brainsci-13-01492-f001:**
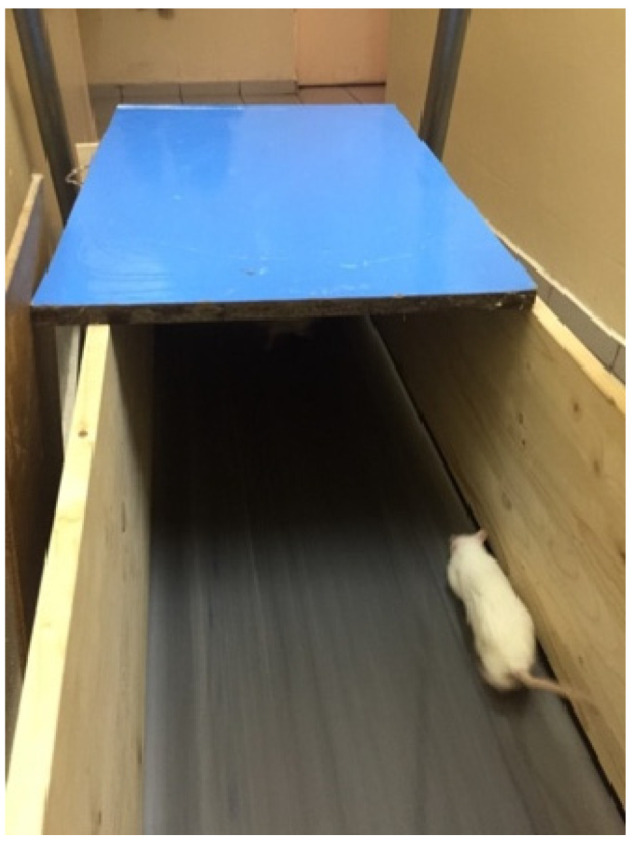
Animal on the treadmill.

**Figure 2 brainsci-13-01492-f002:**
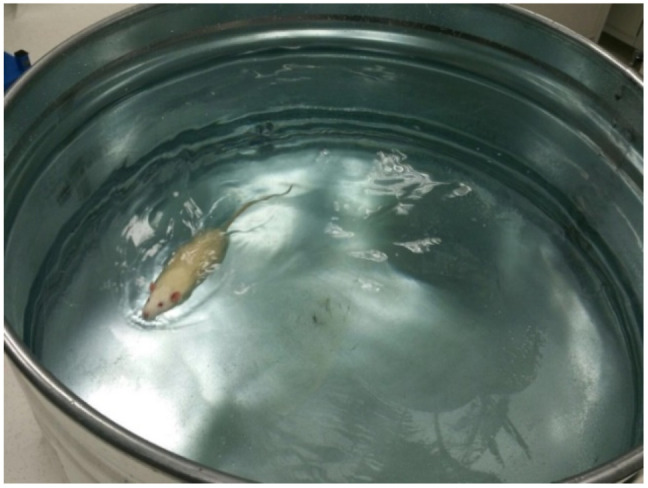
The swimming training.

**Figure 3 brainsci-13-01492-f003:**
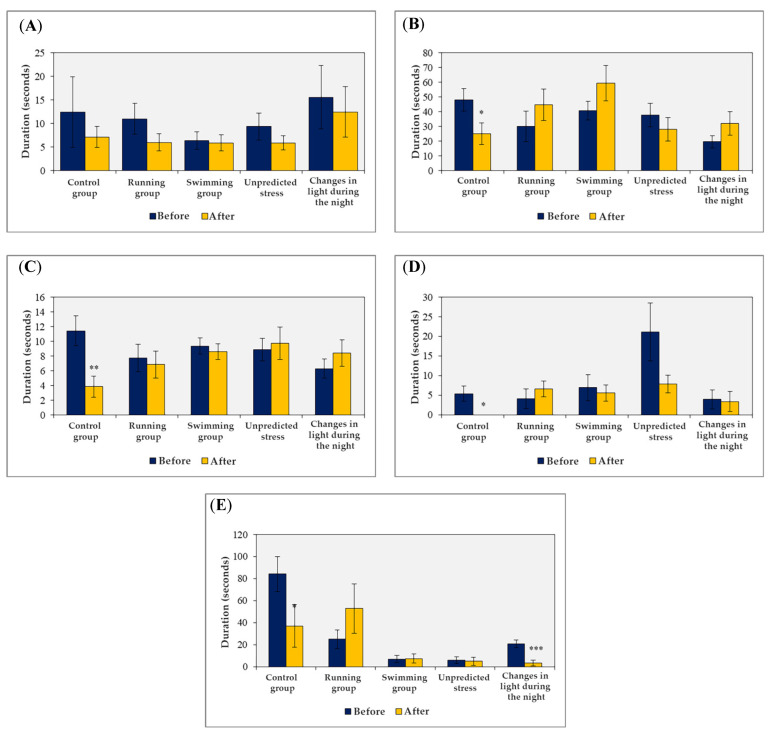
Changes in behavioral parameters observed in the open field test before and after the application of stressors ((**A**) time spent in the central square, (**B**) the number of crossings, (**C**) stretching behavior, (**D**) fur cleaning behavior, and (**E**) freezing behavior). * *p* < 0.05, ** *p* < 0.01, *** *p* < 0.0015.

**Figure 4 brainsci-13-01492-f004:**
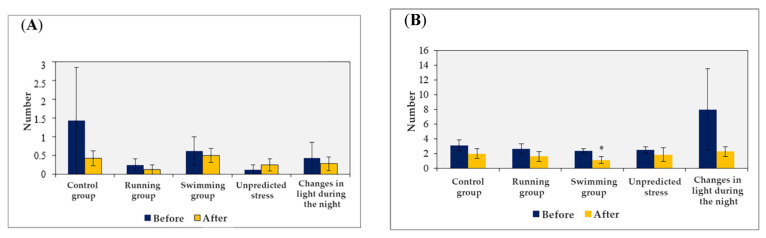
Variations of specific physiological, behavioral parameters, including urine traces (**A**) and fecal boluses (**B**), were observed in the open field test before and after applying stressors. Statistical analysis using the *t*-test assuming equal variance revealed significant differences (* *p* < 0.05) between the two sets of data.

**Figure 5 brainsci-13-01492-f005:**
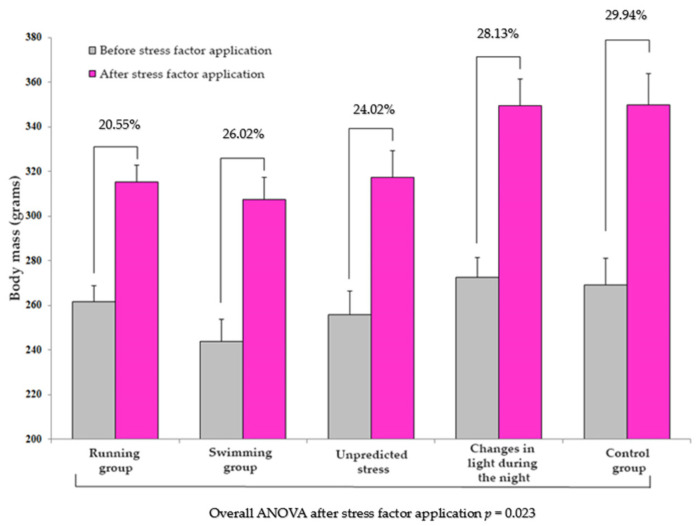
Analysis of weight dynamics in relation to the application of the stress factor. The results are presented as the mean ± SEM, with a sample size of 8 animals per group. The percentage values indicate the growth rate observed in each group calculated by comparing the body mass before applying the stressor to the body mass after the stress exposure period.

**Figure 6 brainsci-13-01492-f006:**
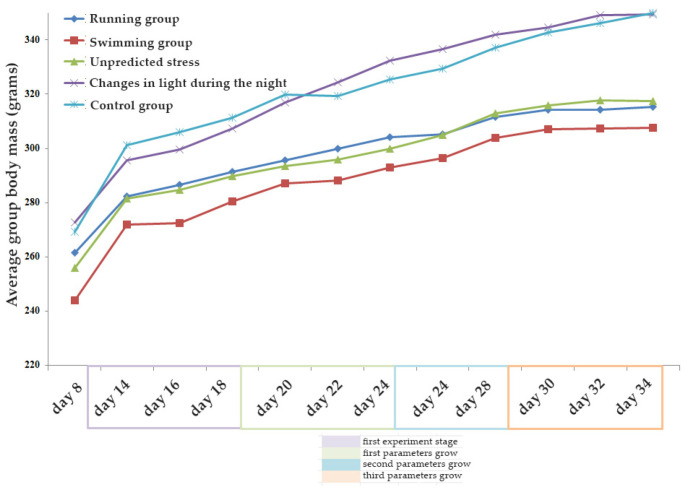
Analysis of body mass dynamics in relation to the application of the stressor throughout the experiment, including key moments such as changes in the intensity of the stressor. The results are presented as the mean ± SEM, with a sample size of 8 animals per group.

**Table 1 brainsci-13-01492-t001:** Distribution of the rats into groups.

Group	Members	Stress Factors
1	1–8	Running
2	9–16	Swimming
3	17–24	Unpredicted stress (alternating running, swimming, light)
4	25–32	Changes in light during the dark phase
5	33–40	Control group

**Table 2 brainsci-13-01492-t002:** Training sessions for groups 1, 2, and 3.

Group	Day 9	Day 10	Day 11	Day 12	Day 13
Running group	1 km/h 7 min	1.5 km/h 7 min	1.5 km/h 10 min	1.5 km/h 15 min	2 km/h 15 min
Swimming group	5 min	8 min	11 min	15 min	15 min
Unpredicted stress	Run 1 km/h 7 min	Swim 7 min	Run 1 km/h 10 min	Swim 15 min	Run 1.5 km/h 10 min

**Table 3 brainsci-13-01492-t003:** Exposure to stress factors.

Groups	Days 14–18	Days 19–23	Days 24–28	Days 29–34
Running group	2 km/h 15 min	3 km/h 15 min	3.5 km/h 15 min	4 km/h 15 min
Swimming group	15 min	25 min	35 min	45 min
Changes in light during the night	Every hour 4 times (22–23, 24–1, 2–3, 4–5)	Every hour 6 times (21, 22–23, 24–1, 2–3, 4–5, 6–7)	Every 30 min—6 times (24–24:30, 1–1:30, 2–2:30, 3–3:30, 4–4:30, 5–5:30)	Every 30 min—8 times (23–23:30, 24–24:30, 1–1:30, 2–2:30, 3–3:30, 4–4:30, 5–5:30, 6–6:30)
Control group	No stressor was applied			

**Table 4 brainsci-13-01492-t004:** Exposure to stress factor of group 3 members.

Day	Activity
14	Swim—15 min
15	Run 2 km/h—10 min
16	The light was alternated on and off every hour during the dark cycle. (22–23, 24–1, 2–3, 4–5)
17	Swim—15 min
18	Run 2 km/h—10 min
19	The light was alternated on and off every hour during the dark cycle. (20–21, 22–23, 24–1, 2–3, 4–5, 6–7)
20	Swim—20 min
21	Run 2.5 km/h—10 min
22	The light was alternated on and off every hour during the dark cycle. (20–21, 22–23, 24–1, 2–3, 4–5, 6–7)
23	Swim—20 min
24	Run 2.5 km/h—10 min
25	The light was alternated on and off every 30 min, 6 times during the dark cycle. (light on from 24 to 24:30, 1 to 1:30, 2 to 2:30, 3:00 to 3:30, 4 to 4:30, 5 to 5:30)
26	Swim—25 min
27	Run 3 km/h—10 min
28	The light was alternated on and off every 30 min, 6 times during the dark cycle. (light on from 24 to 24:30, 1 to 1:30, 2 to 2:30, 3:00 to 3:30, 4 to 4:30, 5 to 5:30)
29	Swim—25 min
30	Run 3 km/h—10 min
31	The light was alternated on and off every 30 min, 8 times during the dark cycle. (light on from 23 to 23:30, 24:00 to 24:30,1 to 1:30, 2 to 2:30, 3:00 to 3:30, 4 to 4:30, 5 to 5:30, 6 to 6:30)
32	Swim—30 min
33	Run 3.5 km/h—10 min
34	The animals were sacrificed

## Data Availability

Not applicable.

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
