# Peer review of "The Impact of Chronic Stress on Behavior and Body Mass in New Animal Models"

_brainsci, 2023, doi:10.3390/brainsci13101492_

Round 1
Reviewer 1 Report
In the manuscript entitled "The impact of stress on behavior and metabolic health", the authors studied the influence of different stress models on the rats behavior, as well as on the change in their body weight.
Although the title gives the impression that it will be about elucidating the impact of stress on metabolism, it soon turns out that this is not the case. Namely, the term metabolism is so broad that body mass only indirectly and very superficially (not even necessarily or directly proportionally) reflects the process(es) of metabolism. So, the first problem of this manuscript is the title which is misleading and suggests that the paper is a literature review, which is not the case. Furthermore, another problem is revealed later, when comes to the material and methods and results sections. Then, namely, one can discover that the previously mentioned connection with metabolism is very indirect. I dare to say that it is too indirect to justify putting in the title of the paper something that will be speculated about in just a few sentences (body weight as a measure of metabolism). In the end, reading the discussion, it come to the point that at least 2/3 of the discussion explains the validity of the applied behavioral tests, and only the rest is directly or indirectly related to what is announced in the title of the paper (the influence of the selected stress on the behavior itself; while metabolic health I discussed earlier). This casts a shadow on the very goal of this study, because the title suggests that the paper will be about the impact of selected stressors on behavior and metabolism, and not the validation of selected stress models through the prism of a specific behavior or a specific behavioral test/model of a mental disorder. That's why I suggest that the authors reconsider the goal of the study and accordingly change the structure of the manuscript so that the title, goals and conclusions have a unique common determinant.
In short, I see the main problem of the study in that the title is not in accordance with the goals that arise from the experimental design and the results, and therefore the discussion. In addition, the objectives are not clearly set out anywhere. All of this together makes the reader confused by the content and somewhat misled by the title.
Apart from these conceptual comments, I would have some other major ones.
In the introduction, there is not a word about the connection of stress with metabolic processes or with "metabolic health", which is a comment that practically continues on what was written earlier.
Line 81: given that you studied the stress-induced behavior of animals, how do you explain the decision to acclimate the animals to laboratory conditions for one week? Also, can you be more specific in that statement since you stated that the animals were acclimatized for AT LEAST one week? Does this mean that perhaps all the animals acclimated longer or that the acclimation lasted differently for different animals?
In line 134, information regarding collecting tears appears and then the same it was done on the last day of the experiment. From the rest of the manuscript, it is not clear why that experiment was done.
Results - Open field test: How do you explain the change in behavior during the open field test that you observed for 4/5 measured parameters before and after the stressor was applied for the control group (which was not exposed to any stressor) and how this affects the interpretation of the results obtained for the groups which were exposed to the stressor and, say, showed the same trend of change?
Lines 306-311: does this paragraph discuss the current study on which the manuscript is based or a previous one? If it is a previous study, references should be given. As far as this study is concerned, this part of the discussion is purely speculation since there is no data on the concentration of melatonin. There is also talk about the pro-oxidative status in the blood, which is linked to leptin, and none of these are supported by the results.
At the end, some minor comments follow.
Line 32, 94: changes in the light of intensity OR changes in the intensity of light?
Line 143/144: missing narrative data for the fourth experimental group (change in lighting conditions)
Line 181: the statement is not clear. It is argued that stress (stress reaction) is important in the response to the stressor.
Line 184: not 32 animals but 40. Control animals are part of the study as well.
Line 213: why is it necessary to emphasize (by contrasting parts of the same sentence) that in some groups there were eight animals and in others the eight also?
Line 291: that many hormones have a rhythmic pattern of secretion dependent on the day-night rhythm is not a hypothesis but a well-established fact.
Line 485: ...an inflammatory. Insult. This process creates... There is a typing error in the sentence.
Author Response
Dear reviewer,
Thank you for all your comments.
We have changed the title according the goal and the results of the study. We mentioned more clearly the objectives of the study in the introduction part.
Line 81 The animals were given a minimum of one week to acclimate before the start of the experiments in order to diminish the errors that can occur in the testing due to the stress that can be achieved by the rats due to the transport. Moreover, the rats needed to accommodate with the new environment, that implied the food, light, humidity and space.
Line 134 We erase this information.
Results - Open field test: We explain the change in behavior during the open field test by the fact that the control group itself became a model of stress through sedentary lifestyle in the absence of cognitive, spatial and social stimuli except for cohabitation. This is the reason why we compare the open field parameters for each group separately before and after the experiment and do not relate them directly to the control group.
Lines 306-311: We erase this information.
Line 32, 94: changes in the intensity of light
Line 143/144: I filled in this information that was missing in table number 3
Line 184: We have made the changes, as you suggested. Thank you!
Line 213: We have added the following: Stress, regardless of its type (light, vigorous physical activity) determines behavioral changes that can be observed in various physiological and psychological disturbances of the study animal. Thank you for your remark!
Line 291: We have corected the by changing the formulation.
Line 485: We have corected this typing error in the sentence. Thank you!
Hope we have touched all the points you asked us to change.
If there are any other changes you consider we should make, please let us know.
Yours sincerely,
All the authors
Reviewer 2 Report
attachment

Author Response
Dear reviewer,
Thank you for all your comments and for observing the positive aspects of our manuscript.
We have changed the title and the introduction in order to highlight the study objectives, as you mentioned. Thank you!
We have added a limitation section, according to your recommendations.
Regarding the ambiguities in the results section, we have added more details. Hope this is all right.
Line 570-574 One of the limitations of our study is the fact that we could not make bioimpedance to the rodents included in our study, in order to make the difference between the changes in their body composition (lean tissue, adipose tissue, water).
Line 597-600
Ambiguity: The mention of "second modification of the stressor parameters" was vague so we have mentioned in the text that the second change in stressors took place on day 24, a detail also found graphically in figure number 6.
Line 623-625 and Line 628-630. This observation was not relevant to the present study so we have erased it.
Regarding the discussion section, we have diminished it. Thank you!
Hope we have touched all the points you asked us to change.
If there are any other changes you consider we should make, please let us know.
Yours sincerely,
All the authors
Round 2
Reviewer 2 Report
I am satisfied with the changes made by the authors, and the manuscript looks good to be published
I am satisfied with the changes made by the authors, and the manuscript looks good to be published.